# Poor Performance of Angiotensin II Enzyme-Linked Immuno-Sorbent Assays in Mostly Hypertensive Cohort Routinely Screened for Primary Aldosteronism

**DOI:** 10.3390/diagnostics12051124

**Published:** 2022-04-30

**Authors:** Agnieszka Łebek-Szatańska, Lucyna Papierska, Piotr Glinicki, Wojciech Zgliczyński

**Affiliations:** Centre of Postgraduate Medical Education, Department of Endocrinology, Bielanski Hospital, 01-809 Warsaw, Poland; piotr.glinicki@wp.pl (P.G.); klinedno@cmkp.edu.pl (W.Z.)

**Keywords:** hormonal hypertension, primary aldosteronism, angiotensin II, aldosterone-to-angiotensin 2 ratio, aldosterone-to-direct renin ratio, enzyme-linked immunosorbent assay

## Abstract

Primary aldosteronism (PA) is the most common, but broadly underdiagnosed, form of hormonal hypertension. To improve screening procedures, current biochemical approaches aim to determine newly appreciated angiotensin II (Ang II) and calculate the aldosterone-to-angiotensin II ratio (AA2R). Thus, the aim of this study was to assess the diagnostic performance of these screening tests in comparison to the aldosterone-to-direct renin ratio (ADRR), which is routinely used. Cheap and available ELISA was used for Ang II measurement. To our knowledge, this is the first study of this laboratory method’s usage in PA. The study cohort included 20 PA patients and 80 controls. Ang II concentrations were comparable between PA and non-PA patients (773.5 vs. 873.2 pg/mL, *p* = 0.23, respectively). The AA2R was statistically significantly higher in PA group when compared with non-PA (0.024 vs. 0.012 ng/dL/pg/mL, *p* < 0.001). However, the diagnostic performance of the AA2R was significantly worse than that of the ADRR (AUROC 0.754 vs. 0.939, *p* < 0.01). The sensitivity and specificity of the AA2R were 70% and 76.2%, respectively. Thus, the AA2R was not effective as a screening tool for PA. Our data provide important arguments in the discussion on the unsatisfactory accuracy of renin–angiotensin system evaluation by recently repeatedly used ELISA tests.

## 1. Introduction

Hypertension is one of the most important, modifiable cardiovascular risk factors [1]. Despite the expanding knowledge and the significant number of available therapies, many patients still present with blood pressure far above the target values [2]. Undiagnosed forms of secondary hypertension, with primary aldosteronism (PA) being the most common [3], constitute one of the reasons for this situation. However, the detection rates of hormonal-dependent hypertension are very low in Europe [4]. Unrealistic screening procedures and uncertainty about interpreting the results of screening tests [5,6] proved to be effectively discouraging to the front-line practitioners. 

The renin–angiotensin–aldosterone system (RAAS) is the key endogenous system in the physiological and pathological regulation of blood pressure and sodium–water balance [7,8]. In PA, aldosterone secretion is uncontrolled and at least partially independent of its physiological regulators. The diagnosis of PA is not easy, as the aldosterone levels may not be elevated [9], which means that the absolute aldosterone excess is not the essence of PA pathophysiology. Thus, renin has proved to be a better marker of relative aldosterone hypersecretion than aldosterone itself [10]. In fact, the most reliable tool for PA detection is the aldosterone-to-renin ratio [9]. However, its usefulness as a screening test is limited by the significant number of required preparations, for example, the withdrawal of interfering antihypertensive drugs. Consequently, the urge to find more convenient or additional tests for primary aldosteronism has led to the exploration of other components of RAAS [11]. This system consists of the numerous angiotensin peptides that arise from the action of angiotensin-converting enzymes in the dynamic cascade [7]. RAAS is a dual system that exerts its physiological effects by two opposite arms: (1) the classic pathway, composed of angiotensin-converting enzyme (ACE), angiotensin II, and AT1 receptor; and (2) the counter-regulatory, non-classic pathway, composed of angiotensin-converting enzyme 2 (ACE2), angiotensin (1–7), and Mas receptor. Angiotensin II is known as the most direct and potent aldosterone regulator. It is sequentially produced first by renin from angiotensinogen and then by ACE through c-terminal cleavage of angiotensin I [12]. Angiotensin II is further converted to angiotensin III, which is believed to also retain the ability to induce aldosterone production [13]. There is little knowledge about the presence and the significance of the non-classic pathway of the RAAS in the human adrenal cortex, although its protective, aldosterone secretion’s blunting role has also been suggested [14].

The determination of the peptides and the proteases of the RAAS may be very useful in practice, as they can serve as novel biomarkers in the diagnosis and monitoring of several pathological states and their treatment. However, the measurement of short proteins such as angiotensins has proved to be technically challenging [15]. Angiotensin II is an octapeptide, with a half-life of 0.5–1 min in circulation [16]. Immunochemical methods constitute the most frequently used laboratory techniques. However, the characteristics and analytical performance of these tests in reference to angiotensin levels determination have not been well established in clinical studies [17]. Due to the limitations of these methods, different techniques based on mass spectrometry have been used for the detection of angiotensins with promising results. As a consequence of studies on angiotensin II [11,18,19,20], some very important advantages of measuring its levels instead of renin in patients with PA have been discussed in the recent literature. In the study of Guo et al. [18], angiotensin II proved to be more sensitive and specific at lower concentrations (as encountered in PA) when compared with renin (both direct renin concentration (DRC) and plasma renin activity (PRA)). Moreover, a smaller percentage of angiotensin II results were below the assay’s functional sensitivity. Regarding the influence of antihypertensive medications on hormonal testing, there have also been important differences between angiotensin II and renin. Both aldosterone and angiotensin II decline as a result of the angiotensin-converting-enzyme inhibitor (ACEI) treatment (in contrast to DRC, which increases in this situation), so the aldosterone-to-angiotensin II ratio (AA2R) is less influenced by these medications than the aldosterone-to-direct renin ratio (ADRR) [19,20]. As illustrated above, the recent data confirms the diagnostic usefulness of the AA2R and its complementarity to the ADRR in certain situations (such as ACEI treatment). In the studies mentioned, angiotensin II was assessed by the newly developed, liquid chromatography with tandem mass spectrometry (LC–MS/MS)-based, equilibrium method. This assay technology allows the quantification of various angiotensin metabolites (equilibrium angiotensins (eqAngs)) simultaneously from one blood sample [21,22,23]. The principle of this method is to establish the ex vivo equilibrated status of the RAAS without blocking any angiotensin-forming or angiotensin-degrading enzymes. Pavo et al. [24] reported a strong correlation between equilibrium and circulating angiotensins (cAng) and between eqAng II and DRC. Similarly, in the study of Basu et al. [25], eqAng II demonstrated a strong, direct dependence on cAng II, both in the population with heart failure and healthy volunteers.

Although LC–MS/MS constitutes the “gold standard” of the angiotensin II evaluation, this method is still hardly accessible and expensive. For screening, cheaper and broadly available tests are necessary. Commercial enzyme-linked immunosorbent assay (ELISA) kits have repeatedly been used to measure angiotensin II levels in studies on COVID-19, cardiology, or even rheumatology [15,26,27,28]. We hypothesized that they can be also useful for PA screening. To our knowledge, there are no reported studies comparing AA2R using these tests with the ADRR in terms of their diagnostic performance in PA. The current study was, therefore, designed to investigate whether AA2R can be used as a screening test and whether AA2R is superior to ADRR in patients receiving their usual antihypertensive therapy, excluding only mineralocorticoid receptor blockers (MRAs). 

## 2. Materials and Methods

### 2.1. Participants

The study was conducted at the Department of Endocrinology Center of Postgraduate Medical Education in Warsaw, Poland. The study cohort included a total of 100 adult hypertensive and/or hypokalemic patients undergoing the diagnostic workup for PA between 2017 and 2020. The study protocol is presented in Figure 1. 

The informed consent was obtained from all participants prior to the first screening test. Exclusion criteria were as follows: (1) pregnancy/lactation, (2) impaired renal function (estimated Glomerular Filtration Rate, eGFR < 60 mL/min/1.73 m^2^), (3) BMI ≥ 40 kg/m^2^, (4) cardiac (Ejection Fraction < 50%) or liver failure, and/or (5) any other serious illness or the aggravation of the chronic disease. Patients were on their usual antihypertensive therapy, excluding only MRAs, as described in detail elsewhere [29].

### 2.2. Hormonal and Biochemical Testing

Blood samples for serum aldosterone, plasma direct renin, and plasma angiotensin II were obtained immediately after admission. After these measurements, the aldosterone/DRC ratio (ADRR) and aldosterone/angiotensin II ratio (AA2R) were calculated. The blood samples were drawn mid-morning (9–12 a.m.), in an upright position, after the patient had been up for at least 2 h and seated for 5–15 min, in accordance with the actual screening guidelines [9]. Basal biochemical parameters, such as sodium, potassium, creatinine (with eGFR calculation), transaminases (ALT, AST), fasting glucose levels, and the lipid profile, were also evaluated. 

After the first screening test (named ADRR I), the diagnostic process continued till the final diagnosis. The patients were prepared for the second test (named ADRR II), including, in particular, the modification of antihypertensive treatment whenever possible and necessary. The results of the ADRR I were taken into account in the decision making of these modifications, which meant, for example, that the discontinuation of renin-lowering drugs was implemented in patients with low renin concentrations, while they were maintained in patients whose renin concentrations were high (despite renin-lowering drugs). Such management complies with the current guidelines. The knowledge of the influence of particular classes of antihypertensive drugs on the results of hormonal tests enables the prevention of the risk of an uncontrolled increase in blood pressure while maintaining diagnostic reliability. 

After ADDR II, diagnostic workup was continued in patients whose results indicated the possibility of primary hyperaldosteronism (according to the current guidelines). Patients with positive and equivocal screening test results underwent confirmatory testing. The seated saline infusion test (SSIT) was used (as described below). Patients with adrenal lesions detected by computed tomography underwent thorough testing for any additional hormonal disturbances (for example, Cushing’s syndrome). Patients with secondary hyperaldosteronism underwent testing for renovascular hypertension. Patients with any form of secondary hypertension other than PA were excluded from the study.

### 2.3. Analytical Details

Blood samples for angiotensin II were collected on EDTA2K, and a protease inhibitor cocktail (PIC) was added immediately, in a proportion of 1:100. The samples were then frozen in low binding protein tubes at −80 °C for less than 6 months. Pre-stored frozen plasma was used to determine angiotensin II manually in a local hospital’s department of radioimmunology with a Human Ang II ELISA Kit (FineTest, Biotech Co., Ltd., Wuhan, China), with no previous chromatographic separation. Analytical sensitivity for angiotensin II was 18.75 pg/mL, and the detection range was 31.25–2000 pg/mL. The intra-assay and inter-assay coefficients of variability (CVs) were <8% and <10%, respectively. The Human Ang II ELISA Kit is based on the competitive ELISA method. A microtiter plate was pre-coated with the target. During the reaction, the target in the sample (or standard) competed with the fixed amount of target on the solid-phase supporter for sites on biotinylated antibodies specific for the detection of the target. After washing the unbound sample (or standard), horseradish peroxidase–streptavidin was added for incubation in each microplate well, followed by the addition of the substrate solution, tetramethylbenzidine, to develop the signal absorbance. The reaction was ended with a sulfuric acid solution, and the color change was measured spectrophotometrically at the wavelength of 450 nm. The intensity of the developed color was inversely proportional to the concentration of Ang II in the samples. The cost of a single test was approximately 4 Euros.

Serum aldosterone and plasma direct renin concentration were assayed using the immunochemiluminescent (CLIA) method on a LIAISON XL analyzer (Diasorin, Vercelli, Italy; CISBio International, Saclay, France). Analytical and functional sensitivity for aldosterone and DRC were 1.45 ng/dL, 1.91 ng/dL, 0.52 μIU/mL and 1.6 µIU/mL, respectively. Inter-assay and intra-assay CVs for aldosterone were <10% and <4.2%, and the measuring range was up to 100 ng/dL. The detection range for DRC was up to 500 pg/mL. 

A seated saline infusion test (SSIT) was performed as follows [30]: patients remained in a seated position for at least 30 min before and during the infusion of 2 L of 0.9% saline intravenously over 4 h, starting at 8–10 a.m. Blood samples for renin, aldosterone, cortisol, and plasma potassium were drawn at time zero and after 4 h, with blood pressure and heart rate monitored throughout the test. Post-infusion plasma aldosterone of >6 ng/dL confirmed PA; the provided plasma cortisol concentration was lower than the value obtained basally (to exclude a confounding ACTH effect). 

### 2.4. Statistical Analysis

The analysis was performed by using STATA 13.1 (Stata Corporation, College Station, TX, USA) and Medcalc 19.7 (MedCalc Software Ltd., Ostend, Belgium). The results are presented as mean ± standard deviation (±SD) for normal data and median (with 25th–75th interquartile range (IQR 25–75)) for non-normal data. A normality test and Bartlett’s test were used to determine the distribution of variables and the equality of variances. Non-normal data were compared using Mann–Whitney-Wilcoxon’s test. Normal data were compared using Student’s *t*-test. The correlations between non-normal variables were assessed using Spearman’s test. The sensitivity and specificity of the evaluated tests were plotted in the receiver operating curves (ROCs). Medcalc 19.7 was used to determine the optimal cut-off points defined as the ones with the highest Youden’s Index. For statistical analyses, DRC values below the assay’s analytical sensitivity limit were rounded up to 0.5 µIU/mL. Angiotensin II values above the assay’s analytical sensitivity limit were rounded up to 2000 pg/mL. A *p*-value of less than 0.05 was considered statistically significant.

## 3. Results

### 3.1. Characteristics of the Study Population

A total of 100 participants, aged 18–70 years, were included in the study. Females constituted 64% and males 34% of the study population. Female and male groups were comparable in terms of age, body mass index, mean systolic blood pressure (SBP), diastolic blood pressure (DBP) values, adrenal lesions found in computed tomography, and the incidence of comorbidities. Men had higher aldosterone and DRC levels (*p* = 0.02 and 0.049, respectively), whereas angiotensin II concentrations were similar in both groups (*p* = 0.33). The basal characteristics of the study population are presented in Table 1.

Hypertension was present in 95% of the study population. Most of them (89%) were treated with the median of two drugs (1–3). Normotensive patients (5% of the study population) were evaluated for unexplained refractory hypokalemia. 

After the final diagnosis, 20 patients were included in the PA group (5 with aldosteronoma and 15 with bilateral adrenal hyperplasia), and 80 in the control group (essential hypertension (EH) group). The groups were comparable in terms of age, sex, body mass index, duration and severity of hypertension, adrenal lesions found in computed tomography, and the number of drugs taken (Table 2 and Table 3).

### 3.2. Angiotensin II, AA2R, and Other Hormonal Results

Angiotensin II levels were comparable between patients with PA and the EH group (773.5 vs. 873.2 pg/mL, *p* = 0.23, respectively) (Figure 2a, Table 4). Moreover, angiotensin II was not correlated with DRC (Figure 2b). Even in the subgroups of patients without ACEI treatment (Figure 2c) and without PA (the EH group, Figure 2d), Spearman’s rho did not improve. AA2R was statistically significantly higher in the PA group than in the EH group (0.024 vs. 0.012 ng/dL/pg/mL, *p* < 0.001). All hormonal test results are presented in Table 4.

The ROCs for angiotensin II and the AA2R ELISA are presented in Figure 3a,b. The optimal cut-off value for AA2R was 0.021 ng/dL/pg/mL. The sensitivity and specificity of this ratio were 70% and 76.2%, respectively (*p* < 0.001). They were statistically significantly lower than the parameters of ADRR I (95% i 87.5%, respectively, *p* < 0.01). The parameters of the test performance of AA2R were similar to those of aldosterone alone (Figure 3c, Table 5).

## 4. Discussion

The current study is the first to report the measurement of Ang II using ELISA in patients with PA in comparison with non-PA/EH. The study aimed to evaluate the practical usefulness of these tests in the screening phase for PA. The patients were intentionally on their usual antihypertensive therapy, excluding only MRAs (as described in detail elsewhere [29]). This decision was due to the fact that the burden of drug withdrawal or modifications is considered one of the factors restraining general or primary care practitioners from PA screening. Therefore, the urgent need for new methods accurate enough in the context of regular patient therapy is very actual.

Due to the complicated nature of the angiotensin cascade, as well as the unstable character and picomolar concentrations of its components, a routine evaluation of angiotensin II (or other angiotensin peptides) was traditionally believed to be problematic. Recently, the newly developed LC–MS/MS-based equilibrium assay technology has been investigated as an accurate method for angiotensin II measurement [18,21,23]. The studies on this method have shown that the AA2R calculated by using equilibrium angiotensin II (eqAng II) represents a promising tool for the diagnostic workup of primary aldosteronism patients. Moreover, AA2R has proved even superior to ADRR or ARR_PRA (aldosterone/PRA) in patients receiving ACEI [19]. However, the access to LC–MS/MS is still strongly limited [31]. The RIA and ELISA assays are the most available and, therefore, the most frequently used laboratory methods to evaluate hormones and endogenous peptides. The main advantages of these methods are lower costs and expertise. However, the specificity of anti-angiotensin antibodies used in these tests may not be sufficient enough to discriminate between small peptides with similar sequences, and this can only be overcome by coupling RIA/ELISA with HPLC [32,33]. Data regarding the evaluation of angiotensin II with these methods in primary aldosteronism is very sparse. Son et al. [34] examined the concentrations of angiotensin II by RIA alone in 111 patients with “idiopathic” primary aldosteronism, 118 patients with aldosteronoma, and 86 patients with EH. Angiotensin II was significantly lower in the PA group than in the EH group—43.2 ng/L (pg/mL) (26.4–74.4) and 60.1 ng/L (pg/mL) (38.5–103.6) versus 56.7 (43.3, 78.9) and 84.3 (61.3, 108.4) in EH (*p* < 0.01), in recumbent and upright position, respectively. There was no significant difference between Ang II concentrations in the “idiopathic” PA and aldosteronoma groups (*p* > 0.05). However, such a difference between these two groups was observed for PRA values, suggesting that PRAs may be more sensitive as markers of aldosterone excess (DRC was not measured). The RIA assay constituted the most reliable and standard method for angiotensin quantification before LC–MS/MS was introduced. The ELISA method has recently been employed for measuring angiotensins in a considerable number of studies; for example, it has recently been extensively used to assess RAS alterations in COVID-19 infections [28]. In the database search, we did not find any studies in which angiotensin II was measured by ELISA in patients diagnosed with primary aldosteronism. In our study, the concentrations of Ang II in the PA group were not statistically significantly different when compared with non-PA/EH group. Moreover, Ang II was not correlated with DRC. Following the suggestions of other authors [18], we examined whether the lack of correlation can be, at least partially, the effect of either low renin levels or the use of drugs. In the study of Guo et al. [18], correlations became weaker in patients with low renin levels (probably due to poor accuracy of renin assays at the lower end of the reference range [35]). Moreover, ACEIs are a class of drugs that cause the opposite effect on DRC (increase) and angiotensin II (decrease), so we hypothesized that the correlation might also become weaker in patients receiving ACEIs. However, we did not observe any differences in the examined subgroups (Figure 3b–d). As studied by Vischer et al. [20], most drugs (except for ACEIs) have similar and unidirectional effects on Ang II and DRC, so we assumed that the use of other kinds of treatment would not influence the results. 

There are several possible explanations why the ELISA angiotensin II measurements and AA2R did not prove their diagnostic value in our study. The pre-analytical phase is probably one of the most important in the whole procedure. The collection and processing of human samples in hospitalized patients or in outpatient settings are very challenging. Therefore, the issue may be not only the test’s performance but simply the fact that measuring circulating angiotensins is very flawed itself. In our study, a commercial protein inhibitor cocktail (PIC) was added to the samples immediately after blood had been drawn. The samples were rapidly transported on ice to the laboratory for further workup. Some researchers [33], however, suggest the opposite order—first adding PIC and then taking blood to the pre-prepared tubes. Although using PIC is not necessary according to ELISA kit producers, in the opinion of experts, it should be recommended [32,36]; the different opinions are less frequent in the literature [27]. Aside from these controversies regarding whether to use PIC or how to use it, there are considerable concerns that even if used, PIC is not potent enough to fully inhibit all protease enzymes present in the samples [21]. The question of whether measuring equilibrium angiotensin levels (which do not require the inhibition of proteases but only the incubation of samples in certain conditions) could significantly improve the poor performance of ELISA kits is yet unanswered. 

Probably the most important problem with the ELISA method is its insufficient specificity and sensitivity to detect picomolar amounts of angiotensins. Most antibodies used by ELISA producers target each peptide’s unique COOH terminus but fail to distinguish the NH2 terminus. This can cause the simultaneous assessment of several angiotensins that share the same COOH terminus with Ang II, such as Ang III and IV. Sample contamination is another significant problem. To address these concerns, Chappel et al. [36] compared Ang II content in deidentified human plasma samples as assessed by two commercial ELISA kits to that obtained from validated RIA. They evaluated Ang II in directly extracted and solid-phase-extracted plasma samples. After the comparison of several specimens, they speculated that ELISA kits detect some unidentified substances in plasma instead of Ang II; substances that are effectively removed by SepPak extraction. Moreover, they observed that, even when samples were solid-phase-extracted, ELISA tests were unable to detect authentic Ang II because their sensitivities were too low. In conclusion, they do not recommend using ELISA for the assessment of Ang II either directly or after solid-phase extraction.

The same concerns emerge from the cited data and our study, which cannot be ignored after the profound analysis of several similar studies using ELISA to assess RAS activity in the different pathological states. Recently, the role of Ang II has widely been debated in COVID-19 as many researchers attempt to evaluate its concentrations in infected patients. Pucci et al. [28] performed a meta-analysis that showed that, in half of these studies, Ang II decreased and in half increased. The increase in Ang II was surprising, considering the fact that in general ACE activity and Ang I levels decreased. The authors also highlight that these observations can at least partially be explained by the poor performance of ELISA tests used in the studies, for example, with the results in the study of Osman et al. [27], who used the same ELISA kit as we used. Studies on RAAS disturbances in systemic sclerosis (SS) have revealed, similar to COVID-19 studies, problematic and opposing results. Although Pignone et al. [37] observed decreased levels of angiotensin II measured by RIA in patients with SS, two studies using ELISA showed increased and unchanged levels of angiotensin II when compared with controls [26,38].

The study of Dai et al. [39] performed in a group of mostly hypertensive patients has presented the same surprising results. In the examined cohort, no correlation between Ang II measured by ELISA and other parameters of RAS, for example, ACE or kallikrein, was present. Additionally, no association between ACE polymorphisms and Ang II levels was found, contrary to the other assessed parameters. In the study of Magalhaes et al. [40], physical exercise did not influence Ang II measured by sandwich ELISA. Conversely, one of the examined exercise protocols significantly increased urinary levels of ACE and plasma levels of ACE2, and the second protocol elevated urinary concentrations of ACE2 and of Angiotensin 1–7; both protocols had no effect on either plasma or urinary Ang II levels. In contrast to these observations, Yang et al. [41] (using the same ELISA kit as Magalhaes et al. [40]) managed to show the differences between plasma Ang II levels in hemodialysis (HD) patients with and without cardiovascular disease, and also between patients on HD and healthy controls. However, no correlations between Ang II and other RAAS components were provided. Moreover, the answer to the question of whether any of these results are reproducible needs further research. 

In conclusion, the presented data confirm that there is a substantial amount of doubt about whether the ELISA method is able to truly detect the differences in Ang II levels. Ang II measured by ELISA is frequently not correlated with other components of RAAS (which was observed for Ang II assessed by LC–MS/MS), and for some Ang II ELISA, results of other potential problems may be present. For example, in the study of Alaagib et al. [42], on a group of non-PA hypertensive patients, Ang II ELISA was near its lower detection limit for most samples. One may conclude that using tests with such sensitivity, the range of results typical for PA patients might be undetectable this way. This is inconsistent with the data that we found from the studies on Ang II measured by LC–MS/MS [18]. As mentioned in the Introduction section, the accurate performance of Angiotensin II LC–MS/MS in measurement near its lower detection limit was one of the main benefits of the assessment of Ang II over renin. In our study, no results were below the lower detection limit using the ELISA Fine Test Kits, but surprisingly again, some of them were above the higher detection limit. Therefore, Ang II ELISA measured with different kits may not perform the same and, therefore, not necessarily display the advantages known for “gold standard” eqAng II LC–MS/MS. Further study is necessary to develop new approaches to immunoassays to guarantee adequate specificity and sensitivity, perhaps on the basis of already proposed solutions [43].

Our study has several limitations, as it was conducted in a single endocrinological center, and the study cohort was relatively small. The ELISA method was performed in plasma samples, which is preferred over serum (as a serum is allowed to clot without the presence of PIC for extended periods of time, causing the falsification of angiotensins content [36]), but with no chromatographic separation or extraction procedures prior to testing. On the other hand, the strength of our study is the thorough radiological and hormonal evaluation of all participants, including all hormonal axes. All patients underwent follow-up, including subtype diagnosis and adrenalectomy, if indicated.

## 5. Conclusions

This is the first study to assess angiotensin II levels using ELISA in primary aldosteronism patients. No statistically significant difference between PA and non-PA groups was observed. Moreover, angiotensin II ELISA was not correlated with direct renin in any of the subgroups analyzed. These observations provide some additional, important arguments to the discussion on the ability of ELISA tests to accurately determine angiotensin II in human samples. This is most significant because these tests have recently been repeatedly used in several studies, with no critical reflection.

The predicted optimal cut-off level of AA2R ELISA was characterized by a sensitivity and specificity of 70% and 76%, respectively. The diagnostic performance of AA2R was significantly poorer when compared with routinely used ADRR, as well as to the parameters of LC–MS/MS AA2R reported recently in the literature. Thus, based on this study’s results, we do not recommend using ELISA tests to evaluate angiotensin II for the AA2R calculation in patients being screened for PA.

## Figures and Tables

**Figure 1 diagnostics-12-01124-f001:**
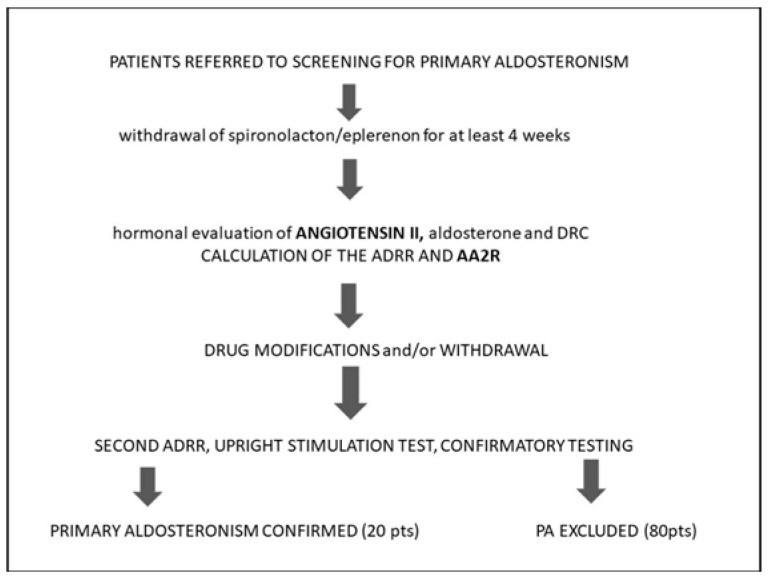
The study protocol. The study cohort included the population referred to screening for primary aldosteronism. Of all antihypertensives, only MRAs were excluded before the first screening test. The participants underwent hormonal evaluation of angiotensin II, aldosterone, and DRC. ARR and AA2R were calculated. After the first screening test, the patients were properly prepared and further diagnosed until the final diagnosis which revealed 20 cases of PA.

**Figure 2 diagnostics-12-01124-f002:**
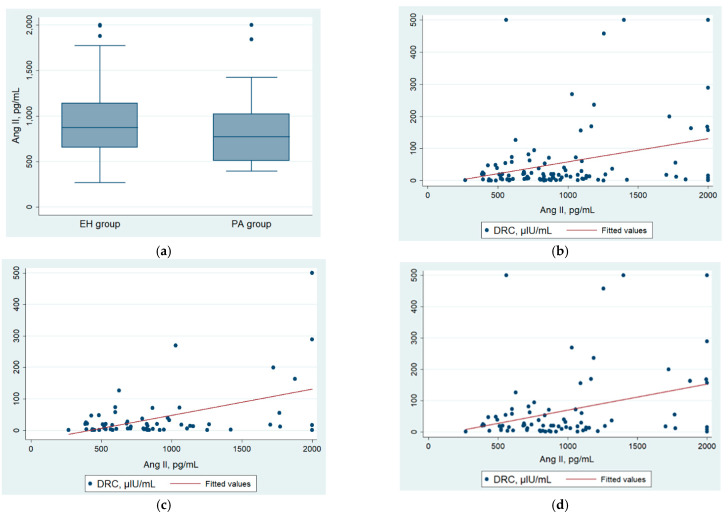
(**a**) Angiotensin II levels were not statistically different between PA (*n* = 20) and EH groups (*n* = 80) (773.5 vs. 873.2 pg/mL, *p* = 0.23); (**b**–**d**) the correlation between Ang II and DRC: (**b**) in the whole study population, with Spearman’s rho = 0.20 (*p* = 0.04), (**c**) in the patients receiving other drugs than ACEI (64 participants) with Spearman’s rho = 0.23 (*p* = 0.07), and (**d**) in the EH group (non-PA, 80 patients), with Spearman’s rho = 0.18 (*p* = 0.12).

**Figure 3 diagnostics-12-01124-f003:**
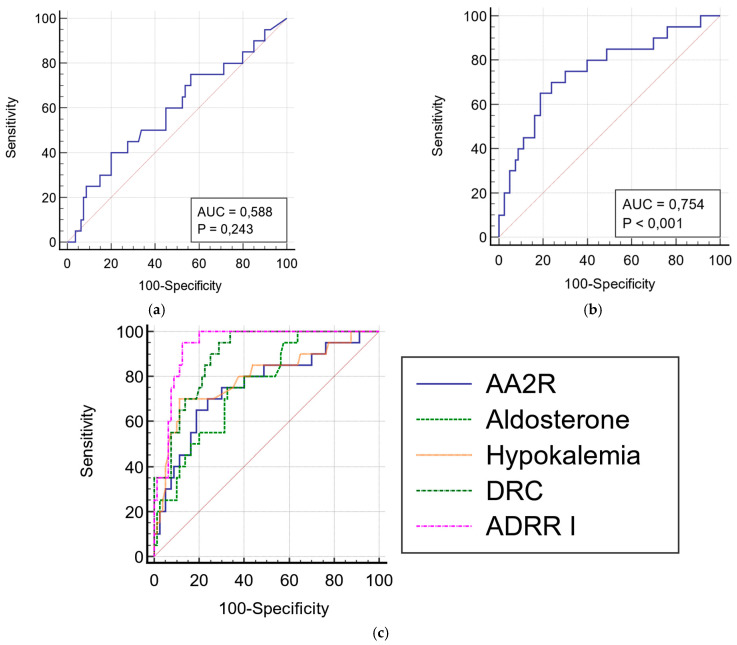
ROCs for (**a**) angiotensin II and (**b**) AA2R; (**c**) comparison of ROCs for ADRR I, AA2R, aldosterone, DRC, and hypokalemia (all *p* < 0.01).

**Table 1 diagnostics-12-01124-t001:** Clinical and biochemical characteristics of the study cohort—comparison between females and males.

Variable	Study Population (*n* = 100)	Females (*n* = 64)	Males (*n* = 36)	*p*
Age (years)	56 (40–63)	58.5 (40–63)	53 (43–62)	0.78
BMI (kg/m^2^)	28.42 (±4.4)	28.04 (±5)	29.25 (±3.3)	0.20
Hypertension (*n*, %)	95 (95)	60 (93.7)	35 (97.2)	0.58
Normotension (*n*, %)	5 (5)	4 (6.25)	1 (2.8)	0.58
SBP (mmHg)	144.95 (±21.6)	144.22 (±23.5)	146.25 (±17.9)	0.65
DBP (mmHg)	89.75 (±12.6)	88.98 (±12.3)	91.11 (±13.2)	0.42
Hypokalemia (*n*, %)	40 (40)	24 (37.5)	16 (44.4)	0.24
spontaneous	24 (24)	12 (18.8)	12 (33.3)	
diuretic-induced	16 (16)	12 (18.8)	4 (11.1)	
Serum sodium (mmol/L)	141.43 (±2)	141.49 (±2.2)	141.31 (±1.8)	0.67
Serum potassium (mmol/L)	4.2 (±0.4)	4.23 (±0.4)	4.14 (±0.4)	0.53
Serum creatinine (mg/dL)	0.76 (0.69–0.9)	0.71 (0.66–0.8)	0.91 (0.76–1)	<0.01
eGFR (mg/mL/1.73 m^2^)	89.23 (79.8–90)	88.39 (80.8–90)	90 (77.8–90)	0.99
Angiotensin II (pg/mL)	852 (600.3–1131.4)	834.4 (543.9–1149.2)	877.9 (713.3–1102.2)	0.33
Aldosterone (ng/dL)	12.2 (8.7–18.8)	11.3 (7–16)	14.15 (10.6–20.1)	0.02
DRC (μIU/mL)	16.13 (3.9–51)	15.07 (3.5–26.5)	27.85 (8–72.2)	0.049

Abbreviations: BMI—body mass index, DRC—direct renin concentration, DBP—diastolic blood pressure, eGFR—estimated glomerular filtration rate, SBP—systolic blood pressure. Data are presented as number and percentage of participants (*n*, %), mean (±SD) for variables with normal distribution, and median (IQR 25–75) for non-normal distributed data.

**Table 2 diagnostics-12-01124-t002:** Clinical and biochemical characteristics of the study cohort—comparison between PA and EH groups.

Variable	Primary Aldosteronism (*n* = 20)	Essential Hypertension (*n* = 80)	*p*
Age (years)	57.5 (46.5–61)	55.5 (40–63)	0.93
Sex (*n*, %F)	11 (55)	53 (66.2)	0.35
BMI (kg/m2)	28.42 (±5.3)	28.42 (±4.6)	0.997
Hypokalemia (*n*, %)	15 (75)	25 (31.2)	<0.001
Spontaneous hypokalemia (*n*, %)	12 (60)	12 (15)	<0.001
Diuretic-induced hypokalemia (*n*, %)	3 (15)	13 (16.2)	0.24
Adrenal lesions (*n*, %)	15 (75)	50 (62.5)	0.43
Serum sodium (mmol/L)	142.24 (±1.9)	141.22 (±2)	0.04
Serum potassium (mmol/L)	3.84 (±0.1)	4.29 (±0.04)	<0.001
Serum creatinine (mg/dL)	0.77 (0.7–0.9)	0.76 (0.69-0.9)	0.46
eGFR (mg/mL/1.73 m^2^)	85.95 (78.8–90)	90 (80.3–90)	0.64

Abbreviations: BMI—body mass index, eGFR—estimated glomerular filtration rate, EH—essential hypertension. Data are presented as number and percentage of participants (*n*, %), mean (±SD) for variables with normal distribution, and median (IQR 25–75) for non-normal distributed data.

**Table 3 diagnostics-12-01124-t003:** Clinical characteristics of the study cohort, excluding participants with no hypertension—comparison between PA and EH groups.

Variable	Study Population (*n* = 95)	Primary Aldosteronism (*n* = 19)	Essential Hypertension (*n* = 76)	*p*
Duration of HTN (years)	7 (2–15)	10 (5–20)	7 (2–13)	0.07
Age of HTN onset (years)	43 (35–55)	42 (38–48)	44.5 (33–55)	0.46
Onset of HTN < 40 years of age (*n*, %)	37 (38.9)	7 (36.8)	30 (39.5)	0.83
Fixed (vs. paroxysmal) HTN (*n*, %)	82 (86.3)	17 (89.5)	65 (85.5)	0.66
Uncontrolled HTN (*n*, %)	66 (69.5)	17 (89.5)	49 (64.5)	0.049
Resistant HTN (*n*, %)	21 (22.1)	3 (15.8)	18 (23.7)	0.55
Mean SBP, mmHg	146.53 (±20.9)	148.95 (±17.2)	145.13 (±22.1)	0.48
Mean DBP, mmHg	90.53 (±12.3)	91.05 (±13.7)	90.39 (±12)	0.84
No treatment (*n*, %)	11 (11.6)	2 (10.5)	9 (11.8)	1.0
No RAAS-interfering drugs—amlodipine only (*n*, %)	13 (13.7) 7 (7.4)	4 (21.1) 2 (10.5)	9 (15.8) 5 (6.6)	0.36
Number of drugs	2 (1–4)	3 (1–4)	2 (2–3)	0.59
Number of drugs in the time of screening	2 (1–3)	2 (1–3)	2 (1–3)	0.83
Modifications of treatment—MRA withdrawal (*n*, %)	34 (35.8) 17 (17.9)	11 (57.9) 5 (26.3)	23 (30.3) 12 (15.8)	0.02 0.32

Abbreviations: HTN—hypertension, DBP—diastolic blood pressure, MRA—mineralocorticoid receptor antagonists, SBP—systolic blood pressure, RAAS—renin–angiotensin–aldosterone system. Data are presented as number and percentage of participants (*n*, %), mean (±SD) for variables with normal distribution, and median (IQ 25–75) for non-normal distributed data.

**Table 4 diagnostics-12-01124-t004:** Hormonal results—comparison between PA and EH groups.

Hormonal Test Results	Study Population (*n* = 100)	PA (*n* = 20)	EH Group (*n* = 80)	*p*
Screening test				
DRC (μIU/mL)	16.13 (3.89–50.98)	2.87 (0.66–5.76)	20.85 (8.97–61.63)	<0.001
Serum aldosterone (ng/dL)	12.2 (8.67–18.75)	18.75 (13.25–24.3)	11.3 (7.27–16)	<0.001
ADRR I (ng/dL/ µIU/mL)	0.64 (0.22–2.62)	5.08 (3.33–20.74)	0.45 (0.2–1.12)	<0.001
Plasma angiotensin II (pg/mL)	852 (600–1131)	773.5 (509.4–1025)	873.2 (653.2–1142.5)	0.23
AA2R (ng/dL/pg/mL)	0.015 (0.008–0.023)	0.024 (0.016–0.032)	0.012 (0.007–0.02)	<0.001
After patients’ preparation				
DRC (µIU/mL)	18.33 (4.84–53.98)	2.53 (0.88–3.91)	24.37 (12.77–71.5)	<0.001
Serum aldosterone (ng/dL)	13.95 (8.83–20.25)	17 (14–25.15)	12.15 (8.31–18.75)	0.003
ADRR II (ng/dL/ µIU/mL)	0.69 (0.22–2.57)	6.03 (3.98–19)	0.48 (0.19–0.98)	<0.001
Confirmatory testing				
Number of participants who underwent SSST (*n*, %)	27 (27)	20 (100)	7 (8.7)	
Post -SSST aldosterone level (ng/dL)	11.6 (5.9–14.8)	14.35 (9.38–17.7)	4.19 (3.4–5.61)	<0.001

Abbreviations: DRC—direct renin concentration, ADRR I—first screening aldosterone-to-direct renin ratio, ADRR II—second ADRR after patients’ preparation, SSST—seated saline suppression test. Data are presented as number and percentage of participants (*n*, %), mean (±SD) for variables with normal distribution, and median (IQR 251–75) for non-normal distributed data.

**Table 5 diagnostics-12-01124-t005:** Comparison of the chosen parameters of ADRR I, AA2R, aldosterone, DRC, and hypokalemia (all *p* < 0.01).

Variable	Associated Criterion	Sensitivity, %	Specificity, %	AUROC	95% CI *
AA2R	0.02 ng/dL/pg/mL	70	76.2	0.754	0.657–0.834
Aldosterone	14 ng/dL	75	67.5	0.757	0.661–0.837
Hypokalemia	3.9 mmol/L	88.7	70	0.794	0.702–0.868
DRC	10.5 µIU/mL	95	71.2	0.892	0.814–0.945
ADRR I	2 ng/dL/µIU/mL	95	87.5	0.939	0.873–0.977

* 95% CI—confidential interval, AUROC—area under ROC.

## Data Availability

The data presented in this study are available on request from the corresponding author. The data are not publicly available due to the privacy of the patients.

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
