# Peer review of "Poor Performance of Angiotensin II Enzyme-Linked Immuno-Sorbent Assays in Mostly Hypertensive Cohort Routinely Screened for Primary Aldosteronism"

_diagnostics, 2022, doi:10.3390/diagnostics12051124_

Round 1

Reviewer 1 Report

Dear authors,

After reading your manuscript I have to say that the subject could be one of interest for other research groups or clinicians.

The scientific data presented in this manuscript is following the literature and the comparison between ELISA and more specific detection/ quantification method of angiotensin II suggested that ELISA is not the best method for screening patients for PA.

You have enrolled 100 adult hypertensive and/or hypokalemic patients undergoing diagnostic work-up for PA. Did you consider in you study protocol any controls like healthy adults for your study? Or this control group is not needed?

However, I believe that the literature needed a study that assess angiotensin II levels by ELISA in patients with PA, hopefully some cheaper and accessible detection methods could be developed by ELISA or other conventional commercial IVD assays.

I appreciate that you have mentioned the study limitations, cause the 100 adult patient cohort provides limited data and I focused most probably on a population that is representative for a small region. However, it works for proving that ELISA method is not sufficient for evaluating angiotensin in PA patients.

There are some other observations:

Line 17 – Typo error “de-termine” to determine.

Line 46 – Typo error “”path-ophysiology” to pathophysiology.

Line 50 – rephrase: “So, the urge to find more convenient… “ – remove “so”or replace it with something else…it sounds not good.

Line 52 – typo error “angioten-sin” to angiotensin. Same error for line 55 “Po-tent”.

Line 63 – replace “so” with something else like hence, therefore, thus….be more scientific. Check the entire manuscript for this.

Line 89: Figure 1 – short description of the study design in figure legend.

Line 170 – “eGFR - estimated glomelular filtration rate” replace with glomerular.

Line 200 - Figure 2. Angiotensin II levels were not statistically different between PA and EH group (773.5 vs 201 873.2 pg/ml, P=0.23). Please add the n value for each group in the figure legend.

Maybe figure 2 and figure 3 could be combined. Same for figure 4 and 5 with ROC curves.

When discussing about the recent literature, regarding the advantages and disadvantages of measuring angiotensin II or renin, or when debating which method to be used, or when discussing the RAAS as key endogenous system in blood pressure equilibrium, cite more research papers. The reference list is short, only 22 sources, it will increase the credibility of the study.

Reference list should be changed and you should follow the guidelines for Diagnostics-MDPI references. See also the recent accepted papers and the guidelines. 

Author Response

Dear authors,

After reading your manuscript I have to say that the subject could be one of interest for other research groups or clinicians.

The scientific data presented in this manuscript is following the literature and the comparison between ELISA and more specific detection/ quantification method of angiotensin II suggested that ELISA is not the best method for screening patients for PA.

You have enrolled 100 adult hypertensive and/or hypokalemic patients undergoing diagnostic work-up for PA. Did you consider in you study protocol any controls like healthy adults for your study? Or this control group is not needed?

Answer: Thank you very much for your review and valuable opinion. We did not consider healthy controls as our work was aimed to evaluate whether ang II and AA2R can distinguish between PA and EH in the screening phase of PA diagnostic process, so we compared this two groups. In the studies on PA diagnostics, EH are in fact the control group. However, from the scientific point of view it would be very interesting to see the differences between angiotensin II in PA, EH and also healthy controls, if the method could prove effective (but it didn’t, unfortunately).

However, I believe that the literature needed a study that assess angiotensin II levels by ELISA in patients with PA, hopefully some cheaper and accessible detection methods could be developed by ELISA or other conventional commercial IVD assays.

I appreciate that you have mentioned the study limitations, cause the 100 adult patient cohort provides limited data and I focused most probably on a population that is representative for a small region. However, it works for proving that ELISA method is not sufficient for evaluating angiotensin in PA patients.

There are some other observations:

Line 17 – Typo error “de-termine” to determine.

Line 46 – Typo error “”path-ophysiology” to pathophysiology.

Line 50 – rephrase: “So, the urge to find more convenient… “ – remove “so”or replace it with something else…it sounds not good.

Line 52 – typo error “angioten-sin” to angiotensin. Same error for line 55 “Po-tent”.

Line 63 – replace “so” with something else like hence, therefore, thus….be more scientific. Check the entire manuscript for this.

Line 89: Figure 1 – short description of the study design in figure legend.

Line 170 – “eGFR - estimated glomelular filtration rate” replace with glomerular.

Line 200 - Figure 2. Angiotensin II levels were not statistically different between PA and EH group (773.5 vs 201 873.2 pg/ml, P=0.23). Please add the n value for each group in the figure legend.

Maybe figure 2 and figure 3 could be combined. Same for figure 4 and 5 with ROC curves.

Answer: Thank you for your thorough review. All the suggested mistakes were corrected, the shortcomings were supplemented and the figures were combined, as was asked.

When discussing about the recent literature, regarding the advantages and disadvantages of measuring angiotensin II or renin, or when debating which method to be used, or when discussing the RAAS as key endogenous system in blood pressure equilibrium, cite more research papers. The reference list is short, only 22 sources, it will increase the credibility of the study.

Answer: The discussion about the recent literature in the Introduction section was enriched with some new information (according to the suggestions of all reviewers) and the references were added. The new reference list is now longer with 43 sources. Thank you for your valuable suggestions.

Reference list should be changed and you should follow the guidelines for Diagnostics-MDPI references. See also the recent accepted papers and the guidelines. 

Answer: The reference list was changed and adopted.

Reviewer 2 Report

To the Authors

General Considerations

The aim of this study was to improve the screening procedures and the current biochemical approaches related to measurement of the angiotensin II (Ang II) using a new cheap ELISA method for the  estimation of the aldosterone-to-angiotensin II ratio (AA2R). Moreover, the results of diagnostic performance of some screening tests were compared to the aldosterone-to-direct renin ratio (ADRR). Authors enrolled 20 patients with primary aldosteronism and 80 controls. The main results of this study were: a) Ang II concentrations were comparable between PA and non-PA patients; b) The AA2R was statistically significantly higher in PA group when compared to non-PA; c) the diagnostic performance of the AA2R was significantly worse than of the ADRR.  Authors concluded that the results of this study add some important arguments in the discussion on the unsatisfactory accuracy of renin-angiotensin system evaluation by recently repeatedly used ELISA tests.

The text manuscript is clear and the results obtained are interesting and in part original. I would like to address to Authors some suggestions in order to further improve the scientific message of the present version of this article.

Specific Points

  1. A very recent article (probably published after the preparation of the manuscript) has reviewed in detail the analytical and clinical aspects of the measurement of biomarkers related to Renin-Angiotensin-Aldosterone and ACE systems (Aimo A. et al. Evaluation of pathophysiological relationships between renin-angiotensin and ACE-ACE2 systems in cardiovascular disorders: from theory to routine clinical practice in patients with heart failure. Crit Rev Clin Lab Sci 2021;58(8):530-545). The conclusions of this review are well in agreement with the data reported in the present study. Authors should cite this review and also compare the results found in this study usig the new ELISA method for Ang II with those discussed in this recent review.
  2. More analytical details should be added about the methods used to measure Aldosterone, Ang II and Renin. In particular, the limit of detection, the between-runs reproducibility (better using the imprecision profile), and more information on the analytical procedure for the Ang II ELISA method (i.e., cost, time of analysis, characteristics of immunoassay system, and so on) should be added in the new version of the manuscript.

Author Response

General Considerations

The aim of this study was to improve the screening procedures and the current biochemical approaches related to measurement of the angiotensin II (Ang II) using a new cheap ELISA method for the  estimation of the aldosterone-to-angiotensin II ratio (AA2R). Moreover, the results of diagnostic performance of some screening tests were compared to the aldosterone-to-direct renin ratio (ADRR). Authors enrolled 20 patients with primary aldosteronism and 80 controls. The main results of this study were: a) Ang II concentrations were comparable between PA and non-PA patients; b) The AA2R was statistically significantly higher in PA group when compared to non-PA; c) the diagnostic performance of the AA2R was significantly worse than of the ADRR.  Authors concluded that the results of this study add some important arguments in the discussion on the unsatisfactory accuracy of renin-angiotensin system evaluation by recently repeatedly used ELISA tests.

The text manuscript is clear and the results obtained are interesting and in part original. I would like to address to Authors some suggestions in order to further improve the scientific message of the present version of this article.

Specific Points

  1. A very recent article (probably published after the preparation of the manuscript) has reviewed in detail the analytical and clinical aspects of the measurement of biomarkers related to Renin-Angiotensin-Aldosterone and ACE systems (Aimo A. et al. Evaluation of pathophysiological relationships between renin-angiotensin and ACE-ACE2 systems in cardiovascular disorders: from theory to routine clinical practice in patients with heart failure. Crit Rev Clin Lab Sci 2021;58(8):530-545). The conclusions of this review are well in agreement with the data reported in the present study. Authors should cite this review and also compare the results found in this study usig the new ELISA method for Ang II with those discussed in this recent review.
  2. More analytical details should be added about the methods used to measure Aldosterone, Ang II and Renin. In particular, the limit of detection, the between-runs reproducibility (better using the imprecision profile), and more information on the analytical procedure for the Ang II ELISA method (i.e., cost, time of analysis, characteristics of immunoassay system, and so on) should be added in the new version of the manuscript.

Answer: Thank you very much for your review and valuable suggestions. We analyzed the suggested article and presented the data from this recent review in the Introduction section. The results of the studies mentioned in the review were discussed in the Discussion. Analytical details have been added in the Methodology section.

Round 2

Reviewer 1 Report

Dear authors, 

Your manuscript was improved and I have no other remarks. 

Congratulations for your paper! I recommend its publication in present form!

All the best!